# Single-Probe Percutaneous Cryoablation with Liquid Nitrogen for the Treatment of T1a Renal Tumors

**DOI:** 10.3390/cancers15215192

**Published:** 2023-10-28

**Authors:** Benjamin Moulin, Tarek Kammoun, Regis Audoual, Stéphane Droupy, Vincent Servois, Paul Meria, Jean paul Beregi, Julien Frandon

**Affiliations:** 1Interventional Radiology Unit, Imaging Department, Institut Curie, 26 rue d’Ulm, 75005 Paris, France; b.moulin00@gmail.com (B.M.); vincent.servois@curie.fr (V.S.); 2Department of Medical Imaging, Nimes University Hospital, University of Montpellier, Medical Imaging Group Nimes, Place du Pr. Robert Debré, 30029 Nîmes, France; tarek.kammoun@chu-nimes.fr (T.K.); jean.paul.beregi@chu-nimes.fr (J.p.B.); 3Urology Department, CHU Caremeau, 30009 Nîmes, France; stephane.droupy@chu-nimes.fr; 4Urology Department, Institut Curie, 26 rue d’Ulm, 75005 Paris, France; paul.meria@curie.fr

**Keywords:** renal cancer, oncology, cryoablation, interventional radiology

## Abstract

**Simple Summary:**

In addressing the challenge of managing small renal masses, this study explored the efficacy of single-probe percutaneous cryoablation as a potential solution. The primary objective was to assess the procedure’s impact on recurrence rates, particularly in relation to Renal nephrometry scores. Analysis of results from 26 renal tumors treated with this method revealed promising outcomes, with recurrence rates influenced by the aforementioned scores. The findings suggest that single-probe cryoablation is a promising modality, particularly when considering its cost-effectiveness and ergonomic advantages over traditional multi-needle argon-based cryotherapy. In conclusion, this novel technique offers potential benefits in treating small renal masses, emphasizing the need for further refinement and research. The study’s insights could guide medical professionals in choosing efficient, cost-effective, and patient-friendly treatment options, thereby benefiting society at large by optimizing kidney tumor management.

**Abstract:**

Kidney cancer accounts for 3% of adult malignancies and is increasingly detected through advanced imaging techniques, highlighting the need for effective treatment strategies. This retrospective study assessed the safety and efficacy of a new single-probe percutaneous cryoablation system using liquid nitrogen for treating T1a renal cancers. From May 2019 to May 2022, 25 consecutive patients from two academic hospitals, with a median age of 64.8 years [IQR 59; 75.5], underwent cryoablation for 26 T1a renal tumors. These tumors had a median size of 25.3 mm [20; 30.7] and a median RENAL nephrometry score, indicating tumor complexity, of 7 [5; 9]. No major complications arose, but three non-clinically relevant perirenal hematomas were detected on post-procedure CT scans. With a median follow-up of 795 days [573; 1020], the primary local control rate at one month stood was 80.8% (21 out of 26). The five recurrent lesions, which exhibited a higher renal score (*p* = 0.016), were treated again using cryoablation, achieving a secondary local control rate of 100%. No patient died, and the disease-free survival rate was 92% (23 out of 25). In conclusion, single-probe percutaneous cryoablation emerges as a promising modality for managing small renal masses. Notably, recurrence rates appear influenced by RENAL nephrometry scores, suggesting a need for further research to refine the technique.

## 1. Introduction

Kidney cancer accounts for 3% of adult malignancies [1,2]. The proliferation of imaging exams, especially computed tomography (CT), has enabled the detection of a significant number of small renal masses. Tumors classified as T1a (measuring less than 4 cm) can either be surgically addressed through partial nephrectomy (PN) or treated percutaneously using thermoablation techniques. These techniques include microwave ablation (MW) and cryoablation (CA), which are often performed under ultrasound (US) or CT guidance [3,4,5,6,7,8]. Compared to PN, CA presents multiple benefits. Notably, CA provides the flexibility to treat all T1a tumors irrespective of their anatomical positioning. Additionally, it is associated with a decrease in complication rates and typically results in a shorter duration of hospital stay [9,10]. Given these advantages, CA emerges as a particularly appealing choice for scenarios demanding nephron-sparing treatments, such as in cases of end-stage renal disease, bilateral tumors, or Von Hippel–Lindau disease. Additionally, it serves as a viable option for patients with multiple coexisting medical conditions [11,12]. Traditional cryoablation devices, which employ high-pressure argon gas, often necessitate the use of multiple needles to achieve effective ablation. This not only prolongs the procedure but also escalates the costs, with the added expense of the requisite argon gas being a significant consideration. Furthermore, the inherent complexities of managing high-pressure gas make these systems cumbersome and less ergonomic. Coupled with the need for a dedicated, often intricate setup in specific operating rooms, these factors limit the mobility and flexibility of conventional cryoablation methods. In a recent development, a new cryoablation system has come to the fore, initially designed for treating breast tumors [13]. Utilizing liquid nitrogen for freezing, this method can produce an ice ball measuring 4.5 × 5 cm using just a single probe, as per the specifications provided by the manufacturer. Moreover, this system employs larger cryoprobes, specifically of 10 and 13 gauges. The safety and effectiveness of this equipment have been documented in a randomized clinical trial focused on the treatment of low-grade breast cancers [13]. This innovation paves the way for potential treatments across a wider range of organs [14,15].

The use of a single probe for cryoablation bears notable advantages. Firstly, it simplifies the procedural complexity associated with conventional methods, streamlining the intervention and potentially reducing the risk of errors linked to multi-probe deployment. Moreover, the utilization of a single probe holds significant economic implications, potentially curtailing the overall costs of the procedure. Furthermore, the single-probe system introduces the potential for creating larger ice balls, yielding the benefit of securing wider safety margins around the targeted lesions. However, while the capacity to create larger ice balls is theoretically promising, its practical clinical implications warrant further exploration and investigation.

The aim of this study is to report on the safety and efficiency of single-probe percutaneous cryoablation with liquid nitrogen for the treatment of small renal masses and to determine the causes of partial tumor response and persistent tumor residue after a T1a renal cryoablation procedure.

## 2. Materials and Methods

This study was conducted in accordance with the Declaration of Helsinki and approved by the Institutional Review Board of Nimes University Hospital, France (protocol code 22.10.02, approved on 12 October 2022). Due to the retrospective nature of this study, which assessed standard care in the participating centers, the requirement for informed consent was waived. However, a letter of non-opposition was sent to participants to ensure transparency and to uphold ethical standards.

All consecutive patients with T1a kidney cancer treated by CA from May 2019 to May 2022 in two university hospitals were retrospectively analyzed from a prospective database. An indication of CA was approved by a multidisciplinary tumor board consisting of a urologist, a medical oncologist, a radiation therapist, and an interventional radiologist (Figure 1). 

### 2.1. Cryoablation Procedure

All patients had an abdominal CT scan or MRI within 6 weeks before the intervention. A biopsy was carried out systematically, prior to discussion at the tumor board. A platelet level ≥50,000/mm^3^, a prothrombin time activity percentage ≥50%, and an international normalized ratio <1.5 were required to realize CA. All CA procedures were performed by senior interventional radiologists (IRs), under general anesthesia. Depending on the tumor location, patients were positioned in the prone, lateral, or supine positions. A liquid nitrogen cryoablation probe (IceSense 3, IceCure Medical Ltd., Caesarea, Israel) was placed under US and/or computed tomography (CT) guidance. Two sizes of probes were used (10 and 13 gauges) depending on the size of the tumor and was left to the IR’s discretion. Two freezing cycles with a passive thaw protocol were applied for treatment. The duration of each freezing cycle depended on the size and the location of the tumor. A CT scan was repeated during the procedure to ensure that the ice ball covered the tumor with the ablation margin. The ice ball volume was measured at the end of the last freezing cycle on US or CT. Final contrast-enhanced CT images were obtained to assess the overall ablation zone and any potential complications. 

### 2.2. Follow-Up

Patients had an MRI or CT and a medical examination with the IR at months 1, 3, 6, and 12 and then each year after the procedure in accordance with the standard care protocols of the centers. When local recurrence occurred, usefulness and/or feasibility of the repeat CA was re-evaluated by the tumor board.

### 2.3. Data Collection

The collected data included patient demographics, tumor histology, prior focal or surgical treatment, presence of a solitary kidney, and number of tumors. For each treated lesion, tumor diameter, tumor volume, ice ball volume, and RENALnephrometry score traducing the complexity of the tumor [16] were also recorded. Complications were collected using the CIRSE classification [17].

### 2.4. Treatment Outcome and Survey Measures

Local recurrence was defined as the presence on follow-up imaging of an enhanced nodule in the ablation zone. In case of uncertain recurrence, a second MRI or CT was performed 2 or 3 months later to evaluate the persistence and/or growth of the nodule. Primary local control was defined as free local recurrence on the treated lesion, after the first session of cryoablation. Secondary local control was defined as free local recurrence on the treated lesion, including patients with a second session of cryoablation. Overall survival (OS) was calculated from the day of the ablation to the time of event, independently to the death origin. Disease-free survival (DFS) was defined as the time from the day of ablation to the day of relapse in any location.

#### Statistical Analysis

Statistical analysis was performed using statistical software SAS and Biostatgv (http://marne.u707.jussieu.fr/biostatgv, accessed on 29 November 2017). Qualitative variables were described using numbers and proportions, and quantitative variables were represented by medians and ranges. Values were compared using the Wilcoxon–Mann–Whitney test. Survival rates are presented using the Kaplan–Meier model and expressed using median survival with interquartile and survival rates with standard error. The ice ball size and the size of the ablation zone were compared, taking the ice ball as reference. Statistical significance was set at *p* < 0.05.

## 3. Results

### 3.1. Demographics

A total of 25 patients (16 men and 9 women, age median 64.8 [59; 75.5] years old) were treated with 26 lesions. The primitive tumor was renal cell carcinoma in 18 cases (69.2%), papillary carcinoma in 5 patients (19.2%), chromophobe carcinoma in 1 patient (3.8%), and oncocytoma in 2 patients (7.7%). The patient’s and tumor’s characteristics are presented in Table 1. 

### 3.2. CA Procedures

A total of 26 CA procedures were carried out (Figure 2). Mean procedure duration was 113.4 [88; 163] min. A 10-gauge probe was used in 14 cases (53.8%) and a 13-gauge probe was used in 12 cases (46.2%), including 12 (46.2%) coaxially inserted. Track embolization was performed in 10 cases, including 7 with resorbable gelatin (26.9%) and 3 with glue (11.5%) (mixture of glue n-butyl-cyanoacrylate and lipiodol (ratio 1:1)). Three CIRSE grade I complications occurred. They were non-clinically relevant peri-renal hematomas detected only on control CT. Despite all adverse events occurring in a patient with a RENAL nephrometry score of 9, complication occurrence was not significantly correlated with RENAL nephrometry score (*p* = 0.07). The majority of patients (n = 24, 92.4%) were discharged at post-operative day 1 (Table 2).

### 3.3. Local Control and Survival

Median follow-up was 795 [573; 1020] days. No nephrectomies were performed after CA. Five lesions in five patients were partially ablated with a primary recurrence confirmed at one month post CA on MRI and/or CT scan so the primary local control was 80.8% (21 out of 26 lesions). The five recurrent lesions, which exhibited a higher RENAL score (*p* = 0.016), were treated again using cryoablation, achieving a secondary local control rate of 100%. The specifics of the five lesions with primary recurrence could be found in Table 3.

Compared to lesions having primary tumor control, lesions with primary recurrence presented higher RENAL nephrometry scores: 9 [8; 9] versus 7 [5; 8] (*p* = 0.016). It was not correlated with tumor long axis (*p* = 0.87), probe size (*p* = 0.68), or index ice ball volume/tumor volume (*p* = 0.87) (Table 4). No correlation was found with a high tumor grade.

There was no alteration in renal function following treatment: creatinine clearance was 51 mL/min [12; 81] before treatment and 49 mL/min [20; 83] 1 month after (*p* = 0.69).

Two patients had metastatic evolution during the follow-up: one with bone recurrence at 200 days post CA and one with controlateral kidney recurrence at 400 days post CA.

No patient died during follow-up. The disease-free survival rate was 92% (23 out of 25 patients) (Figure 3).

## 4. Discussion

This study presents the outcomes of percutaneous cryoablation using liquid nitrogen in treating small kidney tumors (T1a). Throughout and post-procedure, no severe complications were observed. A minor complication identified was the presence of small peri-renal hematomas on post-operative CT scans in three patients (11.5%). The primary local control rate was 80.8%, with five lesions showing early recurrence. These recurrent lesions were successfully treated with a subsequent round of cryotherapy, resulting in a 100% secondary local control rate.

Given the substantial size of the cryoprobe (10 and 13 gauges), many practitioners have voiced concerns about potential complications such as hemorrhage or urinoma, theorizing an increased risk. However, the safety outcomes from our study are reassuring, with complication rates mirroring those cited in existing literature, approximately 8.3% [9]. Detected hematomas in our study were solely identified on CT, and none presented clinical relevance or necessitated transfusion. In a prior study involving 22 patients treated across various organs using the same device, no significant complications were reported, particularly concerning bleeding [14]. Some operators, due to anticipated risks, opted to embolize the puncture path utilizing resorbable gelatin or glue. Although our study does not provide conclusive evidence on the clinical benefits of track embolization, the absence of any clinically pertinent adverse incidents is notable. All complications in our research emerged in lesions with a RENAL score of 9, but this was not statistically significant (*p* = 0.07), likely due to the limited patient sample size. Interestingly, other studies have echoed our findings, establishing a link between a higher RENAL nephrometry score and the onset of complications [18,19,20].

Surgical experts often highlight the RENAL nephrometry score as a reliable predictor for determining the choice between partial and total nephrectomy [16]. Similarly, in our cryotherapy research, the RENAL nephrometry score had correlations with primary tumor recurrence/residue (*p* = 0.016), mainly attributable to the intricacies of tumor location. Tumors that were endophytic and close to the collecting system often posed challenges, complicating the visualization of tumor margins and introducing risks of vascular or urinary puncture. This relationship has been confirmed in other studies [18,21]. Dahlkamp et al. have even proposed that tumors with a high intervention complexity, reflected by a RENAL score of ≥8, should lead to considerations of total nephrectomy [22]. Notably, in our study, all tumors showing residue had a RENAL score of ≥8. A significant advantage of cryotherapy, as revealed in our study, is the ability to retreat tumors that initially resist treatment, achieving a subsequent 100% response rate and preserving the kidney—a major benefit in nephron-sparing approaches.

While the introduction of single-probe cryoablation technology presents notable advantages, it also brings a distinct limitation: the crucial need for optimal needle positioning during the procedure. Unlike conventional methods that permit adjustments through multiple needles, the single-probe system does not allow for such flexibility in real-time corrections. Consequently, there is an emphasized importance on the precision of the initial needle placement. The challenge here is that an imprecise placement could jeopardize both the efficacy and safety of the treatment, especially with complex tumors. For such cases, we advocate the use of CT contrast controls combined with the jet ventilation method. Regardless, if complications arise, a second cryoablation session remains a viable option, ensuring commendable secondary tumor control.

Our study’s oncological outcomes are encouraging. With an average follow-up of 795 days, we achieved a 100% local recurrence-free rate. These findings align with various studies that attest to cryoablation as an effective treatment for T1a kidney cancer. One review reported recurrence rates between 1.5% and 13% for CA-treated kidney tumors, although it noted variations in tumor sizes (that could go beyond 4 cm), CA techniques, follow-up durations, and patient selection [23]. More recently, Andrews et al. highlighted a five-year local recurrence-free rate of 95.9% for all T1a tumors treated with CA [24]. They also noted a 93.4% rate for documented RCC patients, finding no significant differences in outcomes, like cancer-specific survival, local recurrence, metastasis, or death, between those undergoing partial nephrectomy or CA.

A comprehensive review by the American Agency for Healthcare Research and Quality, based on 147 studies, suggested that while PN patients had a slightly lower local recurrence rate than CA patients, incorporating CA retreatment brought their efficacy rates closer [25]. Moreover, CA patients benefitted from reduced blood loss, transfusion rates, conversion to open surgery, and hospital stays. The preservation of renal function was similar for both groups. In our study, there was no alteration in renal function following treatment, as previously described in the literature, similar to partial nephrectomy [23,24,25,26,27]. In line with these findings, our study observed that 19% of our patients required a second CA session to manage residual disease, slightly higher than the 93.4% local recurrence-free survival rate noted by Andrews et al. [24]. The higher recurrence observed in our study might be attributed to the increased presence of complex tumors within our patient cohort. Additionally, the monoprobe approach demands higher ballistic precision to ensure adequate margins and, consequently, to reduce the risk of recurrence. Still, our long-term follow-up showed no recurrence. When comparing complications, our data confirm CA’s minimally invasive nature. A rigorous randomized control trial comparing CA and PN would shed clearer light on their oncological results, complication rates, and broader outcomes.

As we explore the intricacies of renal cryoablation using monoprobe technology, it is clear that future research should also consider the treatment of larger renal tumors, specifically those classified as T1b (exceeding 4 cm). Treating such tumors with a monoprobe presents unique challenges due to their size and complexity. In situations where the monoprobe might not be the optimal choice, other tactics warrant consideration. One such strategy is the pre-cryotherapy tumor embolization [28,29], which may diminish the cold sink effect [30], thereby potentially enhancing the efficacy of cryoablation for more sizable tumors. Moreover, utilizing multiple cryotherapy machines simultaneously could be a strategy to ensure comprehensive and effective tumor ablations. These methods offer potential solutions to the challenges posed by larger tumors, leveraging the benefits of cryoablation technology. Adding to these possibilities is the option of undertaking repetitive treatments which seems possible and safe with this technology. Further studies are crucial to fully realize the versatility of cryoablation for a broader spectrum of tumor dimensions.

The authors recognize several limitations inherent to this study. First and foremost, the absence of a control group makes it challenging to draw definitive comparative conclusions. Coupled with this, the study’s small sample size potentially limits the generalizability of our findings to a broader patient population. The retrospective nature of our research might also introduce biases, as data collection was not originally intended for the study’s objectives, potentially affecting the consistency and comprehensiveness of the gathered information. Moving forward, larger-scale prospective studies with control groups would provide a more robust foundation for affirming the effectiveness and safety of the techniques in question. Despite these limitations, the study sheds light on promising avenues in renal cryoablation, setting the stage for more comprehensive research in the future.

## 5. Conclusions

In the evolving landscape of renal tumor management, single-probe percutaneous cryoablation has showcased its potential as an effective and viable option for treating small renal masses. Our findings underscore the significance of the RENALnephrometry score, with recurrence rates seemingly influenced by these scores. This observation not only sheds light on the complexities of tumor location and configuration but also accentuates the importance of precision in procedural execution. As medical professionals continue to strive for optimal patient outcomes, it becomes imperative to delve deeper into refining and optimizing the cryoablation technique. Future research should prioritize addressing these nuances, with an emphasis on enhancing the procedure’s safety and efficacy across varied tumor complexities.

## Figures and Tables

**Figure 1 cancers-15-05192-f001:**
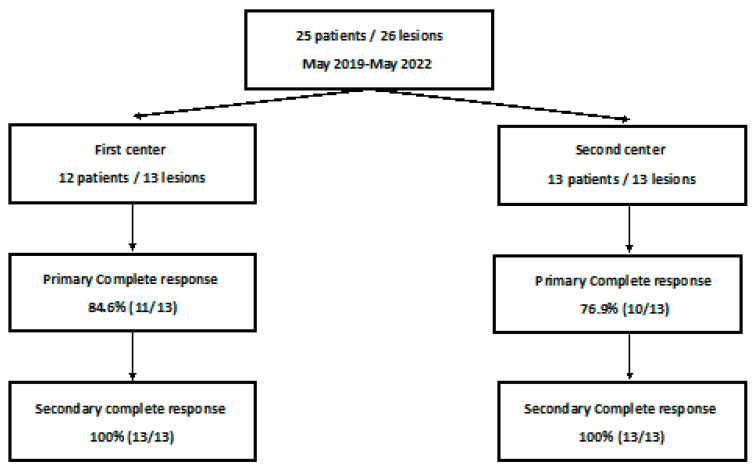
Flow diagram of the study.

**Figure 2 cancers-15-05192-f002:**
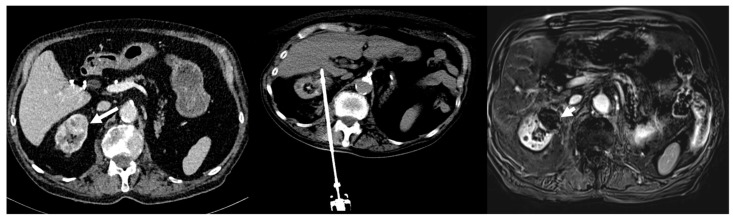
Eighty-five-year-old man with 3 cm biopsy-proven clear cell renal carcinoma (image left, white arrow)). RENAL nephrometry score calculated as 6. The central image shows the CT scan obtained with the patient in a prone position and the cryoprobe placed in the tumor with the final ice ball (end of the second freezing cycle). The right image shows the MRI during follow-up at 1 month, demonstrating complete local response with no evidence of tumor enhancement (white arrow head).

**Figure 3 cancers-15-05192-f003:**
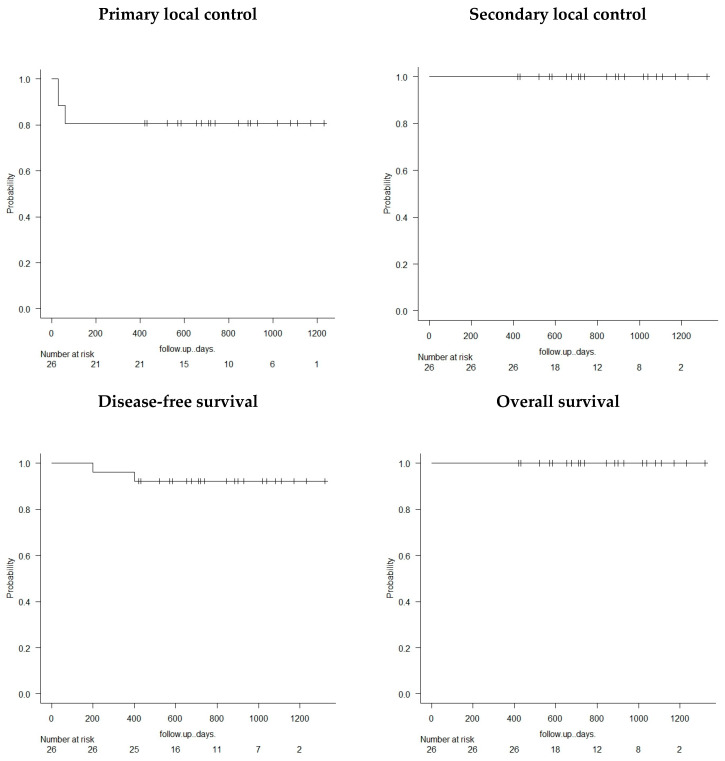
Kaplan–Meier survival curves.

**Table 1 cancers-15-05192-t001:** Patient’s demographics.

Age (Years) (Median [IQR])	64.8 [59; 75.5]
Treated tumors, N (%)	
First center	13 (50%)
Second center	13 (50%)
Sex, N(%)	
Men	16 (64%)
Women	9 (36%)
Histology, N (%)	
Renal cell carcinoma	18 (69.2%)
Fuhrman grading:	
Grade 1	9
Grade 2	8
Grade 3	1
Grade 4	0
Papillary Renal Cell Carcinoma	5 (19.2%)
Type 1	2
Type 2	3
Chromophobe carcinoma Grade 1	1 (3.8%)
Oncocytoma	2 (7.7%)
Creatinin clearance before treatment (mL/min)	51 [12; 81]
Dialysis patients	8 (32%)
Tumor largest diameter (mm) (median [IQR])	25.3 [20; 30.7]
Tumor volume (cm^3^) (median [IQR])	9.9 [4.2; 15.2]
RENAL nephrometry score, N (%)	
low (4–6)	9 (34.6%)
Intermediate (7–9)	14 (53.8%)
High (10–12)	3 (11.5%)

**Table 2 cancers-15-05192-t002:** Cryoablation procedures.

Duration (Minutes) (Median [IQR])	113.4 [88; 163]
Probe size, N (%)	
10 gauges	14 (53.8%)
13 gauges	12(46.2%)
Coaxial use, N (%)	12 (46.2%)
Ice ball volume (cm^3^) (median [IQR])	24 [15.5; 32.6]
Track embolization, N (%)	
none	16 (61.5%)
resorbable gelatin	7 (26.9%)
glue	3 (11.5%)
Adverse events (CIRSE grading), N (%)	
Grade I	3 (11.5%)
Grade II–V	0 (0%)
Post intervention night staying, N (%)	
n = 0 (outpatient procedure)	1 (3.8%)
n = 1	24 (92.4%)
n = 2	1 (3.8%)
Creatinin clearance 1 month after treatment (mL/min)	49 [20; 83]

**Table 3 cancers-15-05192-t003:** Primary recurrence tumors’ characteristics (CCR: clear cell carcinoma).

Patients	Index Ice Ball Volume/Tumor Volume	Freezing Duration (min)	Probe Size	RENAL Score	Tumor Long Axis (mm)	Histology
1	1	24	10 G	9	29	RCC G1
2	1.7	22	10 G	8	24	RCC G2
3	1	20	13 G	11	31	RCC G1
4	2.1	18	13 G	9	27	RCC G1
5	9.3	12	13 G	8	16	RCC G1

**Table 4 cancers-15-05192-t004:** Univariate analysis of primary tumor recurrence (CCR: clear cell carcinoma).

	Primary Tumor Control (n = 21)	Primary Tumor Reurrence (n = 5)	*p* Value
CCR, N	13	5	0.48
RENAL score (median [IQR])	7 [5; 8]	9 [8; 9]	0.016
RENAL score ≥ 8, N	5	5	0.09
Tumor long axis (mm) (median [IQR])	28 [20; 31]	27 [24; 29]	0.87
Index volume ice ball/tumor volume (median [IQR])	3.1 [1.9; 6.1]	2.1 [2; 2.2]	0.87
Probe 10 G, N	12	2	1
Probe 13 G, N	9	3	0.68

## Data Availability

The data are available upon reasonable request.

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
