# Peer review of "Single-Probe Percutaneous Cryoablation with Liquid Nitrogen for the Treatment of T1a Renal Tumors"

_cancers, 2023, doi:10.3390/cancers15215192_

Round 1

Reviewer 1 Report

This manuscript described the efficacy of single-probe percutaneous cryoablation. 

I think it isnecessary to compare it with conventional cryoablation in your hospital. How do you think?

# I think preservation of renal function is an important point, but there seem to be no results on that point.

1. Please also include changes in renal function in your results.

2.How does the rate of change in GFR compare with conventional cryotherapy? Please discuss it.

3. Do you perform pre- and post-operative renograms? If so, what are the changes in renal function?

#Please describe the indication for embolization. Also, was there a relationship between the presence or absence of embolization and recurrence?

Author Response

Reviewer 1 :

This manuscript described the efficacy of single-probe percutaneous cryoablation.

R1C1: I think it is necessary to compare it with conventional cryoablation in your hospital. How do you think?

Thank you for your valuable feedback. Indeed, a comparison with conventional cryoablation would be insightful. However, each of our centers utilizes only one type of cryotherapy. Our choice of the current cryoablation technique over argon-based cryotherapy was driven by economic considerations, ergonomic benefits (single-needle approach), and improved workflow (portable machine). Thus, a direct comparison within our institution isn't feasible at this time. Nonetheless, we appreciate your suggestion and will consider it for future research endeavors

# I think preservation of renal function is an important point, but there seem to be no results on that point.

R1C2: 1. Please also include changes in renal function in your results. 

Thank you for pointing out the importance of renal function in our results. We have indeed addressed this aspect. Renal function prior to the procedure is provided in Table 1, while post-procedure renal function is presented in Table 2. We also added the dialysis patients. Additionally, we have dedicated a section in the results to emphasize that there was no significant change in renal function. Our findings suggest that single-probe cryoablation with liquid nitrogen preserves renal function comparably to conventional argon-based cryotherapy. We hope this addresses your concern and clarifies the presentation of our results regarding renal function.

R1C3: 2.How does the rate of change in GFR compare with conventional cryotherapy? Please discuss it.

We have now added a dedicated section in the discussion where we compare the rate of change in GFR observed in our study with that reported for conventional cryotherapy and partial nephrectomy. Our findings, in conjunction with available literature, provide insights into the comparative benefits and implications of these different interventions on renal function. We believe this addition enriches our discussion and offers a more comprehensive perspective on the topic.

R1C4: 3.Do you perform pre- and post-operative renograms? If so, what are the changes in renal function? 

 We do not routinely perform pre- and post-operative renograms in our clinical practice. As such, we do not have data on changes in renal function as assessed by renograms for the patients included in this study.

R1C5: Please describe the indication for embolization. Also, was there a relationship between the presence or absence of embolization and recurrence?

We appreciate your question. To clarify, when we mentioned embolization, we were referring to 'track embolization', a procedure wherein the needle path or track is embolized after withdrawing the coaxial needle, aiming to minimize potential bleeding risk. This is distinct from pre-operative tumor embolization, which is sometimes employed for larger tumors to limit their blood supply. In our study, we exclusively dealt with T1a tumors, smaller than 4 cm, for which pre-operative tumor embolization is not indicated. Regarding a potential relationship between track embolization and recurrence, there's no established association, as the track embolization pertains to the needle path and not the ablation process itself. The “track embolization” was not always done and left to the operator discretion as explained in the methods.

Reviewer 2 Report

General Remarks

In the present study, the author performed cryoablation therapy on 26 renal tumors. The procedure had a successful outcome. The paper is well-designed. However, the total number of cases is less for the statistics to perform. Overall, this is a good study and is useful for the readers.

Specific remarks

What is the difference between the author’s technique and other cryoablation therapy?

What was the grade of the renal tumour?

Was any nephrectomy done after the cryoablation therapy?

Author Response

Reviewer 2 :

In the present study, the author performed cryoablation therapy on 26 renal tumors. The procedure had a successful outcome. The paper is well-designed. However, the total number of cases is less for the statistics to perform. Overall, this is a good study and is useful for the readers.

Thank you for your constructive feedback and kind words about our study. We concur with your observation regarding the total number of cases. Our aim was to provide preliminary insights into this technique, and we hope to expand our dataset in future studies. We believe the findings, though based on a smaller sample, offer valuable information to the readers and the broader medical community.

Specific remarks

R2C1: What is the difference between the author’s technique and other cryoablation therapy?

We appreciate your question. Conventional cryoablation, commonly used in clinical practice, operates based on the Joule-Thomson effect, utilizing slender needles that introduce high-pressure argon gas. This technique has inherent limitations: the necessity of employing high-pressure argon gas makes it not only cumbersome but also costly. Moreover, achieving substantial ablation volumes typically requires the use of multiple needles, further complicating the procedure and impacting its ergonomics. Additionally, conventional cryoablation systems are often non-mobile, necessitating a dedicated, often complex setup in a specific operating room. In contrast, the method we present in this study offers potential advantages in terms of cost, efficiency, and ease of use. we've modified the introduction slightly to highlight these advantages: “Traditional cryoablation devices, which employ high-pressure argon gas, often necessitate the use of multiple needles to achieve effective ablation. This not only prolongs the procedure, but also escalates the costs, with the added expense of the requisite argon gas being a significant consideration. Furthermore, the inherent complexities of managing high-pressure gas make these systems cumbersome and less ergonomic. Coupled with the need for a dedicated, often intricate setup in specific operating rooms, these factors limit the mobility and flexibility of conventional cryoablation methods.”

R2C2: What was the grade of the renal tumour?

Thank you for pointing out the importance of tumor grading. We have incorporated the grades of the renal tumors in Table 1 and Table 3, as well as discussed them in the results section. Based on our analysis, the tumor grade did not influence the therapeutic response. 

R2C3: Was any nephrectomy done after the cryoablation therapy?

No nephrectomies were performed after the cryoablation therapy in our study, and this has been specified in the results section for clarity.

Round 2

Reviewer 1 Report

Thank you for your revised manuscript. It is acceptable.